# Stem Cells Regenerating the Craniofacial Skeleton: Current State-Of-The-Art and Future Directions

**DOI:** 10.3390/jcm9103307

**Published:** 2020-10-15

**Authors:** Jeremie D. Oliver, Wasila Madhoun, Emily M. Graham, Russell Hendrycks, Maranda Renouard, Michael S. Hu

**Affiliations:** 1Department of Biomedical Engineering, University of Utah, Salt Lake City, UT 84112, USA; 2School of Dentistry, School of Medicine, School of Pharmacy, University of Utah Health, Salt Lake City, UT 84112, USA; emily.graham@hsc.utah.edu (E.M.G.); russell.hendrycks@hsc.utah.edu (R.H.); maranda.renouard@pharm.utah.edu (M.R.); 3Joan C. Edwards School of Medicine, Marshall University, Huntington, WV 25755, USA; madhoun@live.marshall.edu; 4Department of Plastic Surgery, University of Pittsburgh School of Medicine, Pittsburgh, PA 15213, USA; hums2@upmc.edu

**Keywords:** stem cell, regenerative medicine, tissue engineering, craniofacial, cleft palate, bone, cartilage, biomaterials, mandible, reconstruction

## Abstract

The craniofacial region comprises the most complex and intricate anatomical structures in the human body. As a result of developmental defects, traumatic injury, or neoplastic tissue formation, the functional and aesthetic intricacies of the face and cranium are often disrupted. While reconstructive techniques have long been innovated in this field, there are crucial limitations to the surgical restoration of craniomaxillofacial form and function. Fortunately, the rise of regenerative medicine and surgery has expanded the possibilities for patients affected with hard and soft tissue deficits, allowing for the controlled engineering and regeneration of patient-specific defects. In particular, stem cell therapy has emerged in recent years as an adjuvant treatment for the targeted regeneration of craniomaxillofacial structures. This review outlines the current state of the art in stem cell therapies utilized for the engineered restoration and regeneration of skeletal defects in the craniofacial region.

## 1. Introduction

The craniomaxillofacial skeleton boasts a foundational, stable, rigid structure for the overlying soft tissues to encapsulate and form the iconic aesthetic features of the human face. A number of events can be detrimental to this structural framework, including developmental anomalies (e.g., cleft lip/alveolus/palate, Pierre Robin sequence, hemifacial microsomia, etc.), traumatic injury (e.g., facial bone fracture, orbital floor blowout, etc.), or neoplastic lesions. In such cases, traditional craniomaxillofacial reconstructive surgical techniques may not adequately address the long-term sequelae of this tissue detriment. Regenerative medicine and tissue engineering constructs have large application in difficult cases for replacing deformed or malformed craniomaxillofacial tissues, particularly bony tissue, as an adjunct to surgical manipulation. Among these adjunctive regenerative therapies are osteogenic growth factors, biomimetic and biocompatible polymeric scaffolds, and targeted cellular therapies, such as pluripotent stem cell lineages.

Regenerative medicine and surgery, coupled with advances in materials science and tissue engineering, form an alliance of emerging interdisciplinary fields that combine the principles of cellular and molecular biology and biomedical engineering to support intrinsic healing and replace or regenerate cells, tissues, or organs, with the restoration of impaired function. These innovative biotechnologies support natural tissue regeneration processes through the use of cells, natural or synthetic scaffolding biomaterials, growth factors, protein replacement therapeutics, genetic engineering, or a combination of these interventions [1]. Stem cells for use in regenerative medicine and surgery are typically isolated, expanded, differentiated ex vivo, seeded onto scaffolds, and reinserted into the defected areas in combination with tissue-specific growth factors to aid their biocompatibility [2]. Mesenchymal stromal cells (MSCs) are the most common cell type due to their ethical acceptance, ease of harvesting, robust proliferative capacity, and ability to give rise to the foundational cells driving craniomaxillofacial structure regeneration, such as osteoblasts, chondroblasts, adipocytes, tenocytes, myoblasts, and stromal cells [3]. MSCs are typically isolated from human bone marrow-derived stem cells (BMSCs) or adipose tissue-derived stem cells (ADSCs). While BMSCs have long been in the spotlight of tissue engineering strategies, ADSCs have risen to become greater in abundance and can be harvested with less patient morbidity through fat harvest procedures (i.e., liposuction) [4]. Furthermore, there has been mounting evidence in recent years showing that induced pluripotent stem cells (iPSCs) are remarkable materials in regenerative medicine, expanding the arsenal of stem cell adjuvant therapies for tissue engineering constructs [5] (see Figure 1). Biopolymer scaffolds and other biomimetic/bioactive materials can aid cellular growth and differentiation by providing a dynamic three-dimensional (3D) framework for cellular attachment, migration, and protection [6]. Many scaffolds have been optimally engineered to degrade in vivo in a controlled mechanism to prevent immunogenicity within the host [7]. The combined application of growth factors with biomimetic natural and synthetic material scaffolds aims to provide the necessary stimuli to promote the differentiation and migration of stem/progenitor cells toward the optimal cell fates essential for tissue healing [8] (see Figure 2).

While regenerative therapies are actively being explored in a variety of applications in reconstructive surgery for the face and cranium, this targeted review analyzes the preclinical and clinical human trials investigating the efficacy of stem cell therapies in regenerating skeletal structures in the craniofacial region.

## 2. Stem Cells in Calvarial Bone Regeneration

Key potentiators of cellular differentiation and migration, MSCs are widely used in the restoration of calvarial defects; however, one potential drawback of this stem cell type is the lower number of MSCs that fully differentiate into osteoblasts [5]. Furthermore, collecting these cells is considered invasive and poses a number of safety concerns [9]. Saito et al. studied the engraftment and use of reverted induced pluripotent stem cells (iPSCs) and their effectiveness in the regeneration of calvarial bone in rats with a haploinsufficiency of the runt-related transcription factor 2 (RUNX2) [10]. RUNX2 is a gene that transcribes proteins that promote osteogenesis and aid in bone and cartilage development. The RUNX2 transcription factor protein was corrected, in addition to iPSC insertion, to help with the regeneration of the calvarial bone. iPSC-derived osteoblasts were transplanted by injection into the defective areas of calvarium, and osteogenesis was evaluated. This study showed that the osteoblasts without the corrected RUNX2 expressed low levels of differentiation markers and that the transplantation resulted in poor regeneration. However, the reverted iPSCs (with corrected RUNX2) did result in an improvement in the abnormal osteoblast differentiation, leading to a better engraftment into the rat calvarial bone but no remarkable effect on bone mineral density.

Another study also looked at the regeneration of bone in critical-sized defects (intracranial periosteum preserved) from a self-assembling peptide nanofiber hydrogel with iPSCs [5]. This study was conducted in search of a replacement therapy for autologous bone grafts for bone repair and regeneration, which come with many of their own complications. The results of this study showed a significantly higher bone volume after 2 and 4 weeks with the nanofiber hydrogel scaffold with iPSCs as opposed to in the control group with only saline. Medullary cavities and numerous capillaries were present histologically after only 2 weeks in the scaffold+iPSC group, whereas only some fibrous tissues, osteoblasts, and new bone tissue were present for the scaffold alone at four weeks. This suggested that the auto-transplantation of osteoprogenitor cells derived from iPSCs combined with a suitable scaffold would be a good therapy for calvarial bone regeneration.

## 3. Stem Cells in Palatal Bone Regeneration

### 3.1. Mesenchymal Stem Cell-Assisted Osteogenic Regeneration in the Palate

Cleft lip, with or without cleft palate (CL/P), results from either the failure of the fusion of facial processes and/or fusion of palatine shelves. CL/P is the most prevalent congenital craniofacial defect, affecting ~1 in 500–700 live births worldwide each year [11]. Treatments require early and repeated reconstructive surgeries by surgical specialists as well as non-surgical specialists including dental, speech pathology, genetic medicine, and nutrition experts. Craniofacial defects negatively affect quality of life and result in increased mental health challenges, as well as causing increased economic burdens on hospitals and patients. Fortunately, bone tissue engineering techniques utilizing the regenerative capacity of stem cells are a promising alternative to the use of invasive surgeries to reconstruct bone defects in the craniofacial region.

Localized human adult stem cell delivery, especially of MSCs, for tissue regeneration is considered a fundamental cell therapy technique due to the cells’ unique ability to self-propagate and differentiate into various cellular phenotypes and has been widely applied to craniomaxillofacial tissue types in tissue engineering. In contrast to the invasive nature of autologous bone grafting for the repair of bony defects, autologous stem cells can be harvested by minimally invasive procedures, including via needle aspiration, with a reduced risk of complications at the donor site. MSCs have the potential to differentiate into osteocytes, chondrocytes, and adipocytes, depending on the growth factors in the microenvironment. MSCs also have the capability to modulate the immune response and promote tissue regeneration [12]. Furthermore, MSCs can be isolated from many tissues, with bone marrow and adipose tissue being the most common adult sources in clinical practice. Other MSC/MSC-like cell populations can be extracted from a variety of organ types, including the skin, pancreas, heart, brain, lung, kidney, cartilage, tendons, and teeth.

Cell-based therapies have been evaluated over the past several decades as potential regenerative interventions to alleviate the burden of alveolar and palatal clefts, as MSCs and other MSC-like cell populations have demonstrated the capacity to efficiently differentiate toward osteoblast lineages, driving the formation of new bone tissue [13]. This capacity has been demonstrated in studies analyzing MSCs together with recombinant human bone morphogenetic protein-2 (rhBMP-2), a potent osteogenesis-potentiating protein molecule, for the treatment of alveolar clefts [14]. While rhBMP-2, and its analogous rhBMP-7, have been FDA-approved for clinical use for alveolar clefts, their combination with cell-based therapeutics such as MSCs remain under investigation for validation with long-term outcomes data. Below, we discuss different cell populations and their preclinical and clinical evaluation in a variety of studies for the treatment of alveolar and palatal clefts.

### 3.2. Bone Marrow-Derived Stem Cells

Bone marrow-derived stem cells (BMSCs) have been frequently investigated and are considered the “gold standard” for bone tissue regeneration. These cells have both positive and negative characteristics that contribute to their potential use in CL/P repair. Compared to other mesenchymal stem cells, stem cells from bone marrow demonstrate increased osteogenic and chondrogenic differentiation potential [12]. However, one study noted that patient biological factors, procedure protocols, and growth factors may affect the differentiation capacities of these mesenchymal stem cell sources. Some drawbacks to using these cells according to Wu et al. in 2019 are their loss of proliferative and differentiation capacities during cell expansion, the increased risk for pathogen contamination and genetic transformation, and the fact that they are negatively affected by increasing age of the donor.

BMSCs are obtained by bone marrow aspiration, which is carried out at the patient’s bedside using local anesthetic at the preferred location of the posterior superior iliac spine, or by in vitro cultivated BMSCs. Bone marrow aspirate has been noted to have higher osteogenic potential compared to cell populations cultivated in vitro [15]. Obtaining these cells is generally a safe procedure, but due to its invasive nature, pain at the donor site is common, as well as an increased risk for surgical infection [16]. Aspirates of BMSCs may also not contain enough cells to induce bone formation and may need in vitro culture expansion to increase the population of cells, which is expensive and time-consuming [17]. Contrarily, adipose-derived stem cells are more readily obtained and can be rapidly expanded.

Using BMSCs in oral and craniofacial surgical bone reconstruction has shown encouraging results. Several in vivo studies investigating maxillary sinus floor elevation using BMSCs showed increased new bone formation compared to traditional methods, as well as the successful application of BMSCs in periodontal intra-bone defect regeneration and jaw defect reconstruction after the enucleation of a cyst. BMSCs have been utilized in several alveolar cleft surgeries and show favorable results. In one of these studies, it was concluded that a scaffold-free approach to BMSC bone marrow stem cell-mediated bony reconstruction is safe for alveolar cleft repair, but not indicated for large cleft deficiencies. There have been a number of associations drawn between BMSC populations and their in vivo activity toward commercially available demineralized bone matrix (DBM), such as Osteoset DBM, and platelet-derived growth factor when used in tandem with tricalcium phosphate/hydroxyapatite or platelet-rich fibrin composites, contributing to bone repair mechanisms [18]. Importantly, however, BMSCs have not been shown to alleviate the morbidity caused by iliac crest autologous harvest, even by way of minimally invasive extraction techniques, necessitating the further investigation of other MSC sources [19,20]. A later study employed a collagen scaffold seeded with autologous bone marrow mononuclear cells, platelet-rich fibrin, and nanohydroxyapatite, showing enhanced healing and decreased post-operative pain compared to the traditional method of cleft palate repair using iliac bone grafting [21]. A current randomized clinical trial is investigating the efficacy of bone formation in unilateral alveolar clefts with bone marrow-derived stem cells on a collagen matrix (Osteovit) compared to iliac crest autograft.

### 3.3. Adipose-Derived Stem Cells

Similar to BMSCs in growth and differentiation potential, adipose-derived stem cells (ADSCs) offer several qualities that make them a promising alternative to BMSCs or traditional (autogenous) bone grafts. Boasting a higher cell-to-volume proportion than other traditional cell lines, ADSCs are less sensitive to aging, compounded by the ease of harvest and application of the isolated stromal vascular fraction, enriched with potent growth factors for improved engraftment outcomes [22].

Along with their ability to directly differentiate into osteoblasts, ADSCs produce chemokines that are useful for facilitating the homing of endogenous stem cells to the site of a bone defect. One of the many challenges for a successful stem cell translation is for the MSCs to survive in the hypoxic conditions that follow graft placement. For proper bone regeneration and healing to occur, angiogenesis to the graft site must occur for the proper exchange of oxygen, nutrients, growth factors, and metabolic wastes. ADSCs are the ideal stem cell therapy due to their ability to survive in hypoxic environments by secreting several growth factors such as vascular endothelial growth factor (VEGF) and platelet-derived growth factor (PDGF) that induce angiogenesis. These pro-angiogenic factors promote blood vessel formation and recruit hematopoietic cells to allow the exchange of oxygen, nutrients, wastes, and growth factors necessary for cell survival [22]. ADSCs, on the other hand, are negatively affected by donor age in their osteogenic potential.

A promising future direction for palate bone regeneration is the use of the stromal vascular fraction (SVF) from human adipose tissue. The SVF is a single source of a diverse population of cells that include multipotent stem cells; progenitor cells, including endothelial cells; stromal cells; pericytes; preadipocytes; hematopoietic stem cells; and macrophages. Due to the ADSCs in the SVF having a variety of cells, they demonstrate osteogenic and angiogenic potential in in vitro observations and in vivo in murine studies. In a clinical trial, a group used the SVF/ADSCs in a maxillary sinus floor elevation to develop vascular and bone formation and showed a higher bone mass that positively correlated with blood vessel formation compared to in the control group [15]. In addition to being in the SVF, ADSCs have shown the ability to release plasma membrane-derived vesicles (MVs) into the microenvironment. These vesicles are important in cell-to-cell communication due to secreting angiogenic molecules such as FGF2, PDGF, VEGF, matrix metalloproteinase (MMP) -2, and MMP9, as well as osteogenic molecules, such as BMP2, and RNAs and microRNAs that impact local neighboring cells as well as having effects throughout the body. Future research using these techniques still needs to be investigated, but early studies comparing adipose tissue-derived microvascular fragments (MF) and the SVF have suggested that MF shows higher angiogenic potential than the SVF. The clinical benefit of using these techniques is that they are ultimately safer for the patient because they contain a variety of cells that can be collected in large amounts in a one-step surgical procedure that decreases the procedure time and chance of infection compared to those with BMSCs, endothelial cells, or autologous bone harvesting [15].

ADSCs are obtained from adipose tissue through liposuction or during reconstructive surgery through the resection of tissue fragments. Several case reports have reported the successful application of ADSCs in the regeneration of craniomaxillofacial bone defects [22]. Buccal fat pad-derived ADSCs were used in alveolar cleft reconstruction by Khojasteh et al. and were cultivated in vitro, seeded on demineralized bovine bone mineral (DBBM) and autologous bone (AB). The results six months after treatment showed more bone development in the test group using ADSCs compared to the control group [15]. There is growing evidence that ADSCs have become optimal candidates for musculoskeletal tissue CL/P repair, given their broad ability to differentiate into various tissue types and their added capacity to expand in vivo [4]. In a pre-clinical in vivo investigation, Pourebrahim et al. applied undifferentiated ADSCs to maxillary alveolar cleft bony defects through vehicular delivery with biphasic bone substitutes, including hydroxyapatite-tricalcium phosphate scaffolds [4]. The authors concluded that this delivery system offered bone regeneration at the site of repair comparable to that with autologous tissue transfer and may be a preferable alternative to autografts given the lack of patient donor site morbidity and operative time. By contrast, Conejero et al. found substantial bone formation when poly-L-lactic acid scaffolds seeded with osteogenically differentiated ADSCs, but not their multipotent progenitors, were implanted into rats [23]. Blanco Elices et al. generated bilaminar fibrin–agarose hydrogels immersed in tranexamic acid-supplemented human plasma containing cultured ADSCs and palate-derived oral mucosa fibroblasts in an attempt to reproduce palatal hard and soft tissues in rabbits [24]. In vivo cell and tissue differentiation could regenerate the palatal tissues—however, not to control levels—following 4 weeks of observation. Novel culturing methods have also been examined for the packaging and differentiating of human stem cells for palatal bone regeneration, including spheroid delivery systems to optimize cell colonization in scaffold materials in vivo [25]. While still in the initial stages of investigation, these data are promising for the implementation of mesenchymal stem cell adjunctive therapies to induce osteogenesis in cleft defects.

### 3.4. Tooth-Derived Stem Cells

Within dental tissues, there are five main classes of MSC populations, including (I) dental pulp stem cells (DPSCs), (II) stem cells from exfoliated deciduous teeth (SHEDs), (III) periodontal ligament stem cells (PDLSCs), (IV) dental follicle progenitor stem cells (DFPSCs), and (V) stem cells from apical papilla (SCAPs) [26]. Beyond the five main classes, neural crest stem cells have also emerged as a prominent cell population arising in the dorsal neural tube that give rise to a diverse array of cell types in the body, including craniofacial cartilage and bone [27]. The most easily extracted and isolated of the aforementioned odontogenic cell lines are the SHEDs, commonly isolated through minimally invasive approaches. Similar in terms of the demonstrably high levels of immune-stimulating and modulating chemokines, broad and diverse differentiation profile, and strong proliferative capacity of the well-studied DPSCs, SHEDs are commonly isolated in young patients aged 5–12 years (as opposed to DPSCs, more commonly isolated from teenage years onwards from third molar tooth-extraction sites) [28]. Additionally, there is great potential in both SHEDs and DPSCs for the generation of cell sheets and cultures using three-dimensional spheroid delivery systems [29,30]. Both, highly concentrated with secretomes (e.g., soluble paracrine signaling molecules), allow for their immunomodulatory, angiogenic, and neurogenic activities in vivo [31]. The unique secretome profile of SHEDs shows modulatory activity during differentiation in osteogenic lineages, leading to an increase in neoangiogenesis [32]. Additionally, SHEDs have been shown to effectively form new calvarial bone in a critical-size defect experiment, with greater capacity compared to other odontogenic tissue-derived cell lines in an FGF-2-primed collagenous hydrogel deprived of oxygen, ultimately demonstrating significantly increased intramembranous ossification capacity [33].

There is much interest in deciphering the clinical applicability of dental tissue-derived stem cells in the regeneration of oral and craniofacial bone structures. An ongoing randomized control trial of secondary alveolar cleft repair is seeking to compare autogenous deciduous dental pulp mesenchymal stem cells seeded onto scaffolds composed of hydroxyapatite/collagen matrices with iliac crest autogenous bone grafts. De Mendonca Costa et al. conducted a study on non-immunosuppressed rats that evaluated the effectiveness of human-derived dental pulp stem cells (hDPSC) in the reconstruction of cranial bone defects [34]. The hDPSCs were characterized in vitro as mesenchymal cells and were evaluated for cell markers that showed osteogenic, adipogenic, and myogenic differentiation. The hDPSCs were integrated within a collagen scaffold and were evaluated alongside a collagen-only group. After one month, both groups showed signs of bone formation; however, the hDPSC rats demonstrated more mature bone. Another significant finding of this study was the lack of signs of graft rejection, suggesting that hDPSCs are another potential cell source for correcting cranial defects. Another study used hDPSCs alongside a silk fibroin scaffold and human amniotic fluid stem cells (hAFSCs) to repair cranial bone defects in immunocompromised rats [35]. This study demonstrated that the hAFSC-seeded scaffold produced higher bone formation than the fibroin scaffold alone after only 4 weeks postoperatively. Finally, Jahanbin et al. tested the effectiveness of hDPSCs on maxillary alveolar defects in rats with a collagen matrix against iliac bone graft transplantation [36]; new bone formation was evaluated 1 and 2 months after surgery. The results of the study showed maximal new bone formation in the iliac bone graft group after 2 months. However, the hDPSCs did show significant bone formation after 2 months compared to the collagen scaffold control group.

## 4. Stem Cells in Craniofacial Cartilage Regeneration

A study from Maruyama et al. suggests that calvarial sutures serve as the growth center for the development of the craniofacial skeleton and are the major site of bone expansion during postnatal craniofacial growth. Cranial neural crest cells (cNCCs), upon stimulation from pro-migratory and pro-osteogenic signaling cascades (e.g., Wnt/Beta catenin), form the primordial proliferating source of developing cells during craniofacial development [37]. It has been postulated that suture mesenchyme is a source of skeletal stem cells [38]. However, the quality of cranial suture stem cells (SuSC) and their mechanism of bone regeneration have been little studied. The authors of this study isolated a SuSC population and tested for its expression of Axin2 (a marker to identify slow-cycling stem cells) and its differentiating abilities. They further tested the abilities of the isolated SuSC to behave like stem cells in skeletal repair. Axin-2-expressing cells showed the ability to differentiate into osteogenic cells and contribute directly to bone regeneration, which further supports that any Axin-2-expressing cells contain adult SuSC. SuSC were also found to give rise to chondrocytes, indicating multipotency; this was tested and manipulated through BMP signaling. Those cells with exogenous BMP2 revealed cartilage but not mineral, therefore altering the commitment of SuSC cells to a chondrogenic lineage. This study concluded that the direct engraftment of SuSC to injured bone provided benefits for cartilage repair, especially through altering BMP signaling, lending this cell population utility in the subsequent formation of intramembranous bone.

Stromal stem cells have also been studied for the repair of auricular cartilage, and some studies indicate their properties in healing craniofacial defects [39]. In the study conducted by Kuroda et al., bone marrow stromal cells were embedded within a collagen scaffold to repair a defect in the medial femoral condyle of an athlete. The results showed that seven months post-transplantation, hyaline-like cartilage tissue and smooth tissue were present. A more recent study showed that human adipose-derived stromal cells (hADSCs) could aid in the treatment of skeletal defects [40]. The main question was if ADSCs could be used pre-differentiation to promote shorter hospital stays and reduce the risk of contamination. This study also found that no additional medium is needed prior to engraftment and that ADSCs are not detected at the site of injury after two weeks. This suggested a clear bone turnover and a stimulation of the host’s reparative processes. This study also mentioned the significance of BMP signaling in modulating hADSC to mediate different types of bone repair.

## 5. Stem Cell-Mediated Mandibular Defect Regeneration

Many different scaffolds are being tested in conjunction with cell-based therapies to provide anatomically precise reconstructions of difficult areas, such as the mandible. Fang et al. conducted a study showing an increase in bone-to-tissue volume, bone thickness, and confirmed bone-implant integration in a critical mandibular defect in a rabbit model using a polyether-ether-ketone (PEEK) scaffold with ADSCs [41]. The PEEK combined with ADSCs also provided an increased compressive resistance, which provides a solution for the forces presented during mastication. Another study investigated MSCs modified with the transcription factor RUNX2 in rabbit mandibular reconstruction and showed that a significantly higher amount of new bone was formed after only 8 weeks [42]. In addition to new bone growth, the quality of the bone was also found to have increased, in terms of mineral density, bone mineral content, and maximum load strength, suggesting that MSCs along with RUNX2 were effective at promoting bone regeneration in rabbit mandibular defects and would therefore be a valuable strategy for craniofacial fracture reconstruction.

A more recent study compared the effectiveness of a different scaffold material, fibrin glue, with ADSCs against the standard autologous bone grafts in large mandibular defects in rabbits [43]. The computerized tomography (CT) results after 8 weeks showed significant cortical bone reconstruction in the “fibrin glue associated with ADSCs” group. A statistically significant difference in the thickness of the new bone of the ADSC group versus the group that was treated with the scaffold alone was also noted; however, no significant difference for thickness was found between the ADSC group and the autologous bone graft group.

Although autologous grafting is effective, it still comes with some disadvantages and is considered a highly invasive approach to reconstruction. A clinical trial conducted by Gjerde et al. evaluated mandibular regeneration using bone-derived MSCs for 11 subjects with severe mandibular ridge resorption [44]. The results were assessed after 12 months; the bone augmentation of the posterior mandibular ridge (one of the more challenging sites for reconstruction) was successful for all 11 patients. This clinical trial was different than any other study in that the MSCs were not manipulated with any growth factors or stimulants. This was to ensure the growth factors did not have different effects on different tissues and to reduce production costs; however, this slowed cell expansion in vitro prior to seeding. Although the clinical trial had a small cohort and further studies and follow-up are necessary, this study showed successful reconstruction and augmentation that compete with the gold standard of autologous grafting. Preliminary clinical studies have shown successful reconstruction with the combination of autologous bone grafts and human BMSCs (followed by distraction osteogenesis, dental implants, and prosthodontic restoration) for a mandibular defect caused by an odontogenic myxoma [45] as well as for a plexiform ameloblastoma of the mandible [46]. Finally, a preclinical murine study of BMSCs cultured for 7 days in an osteogenic medium (for extracellular matrix deposition) prior to implantation demonstrated the heterotopic ossification potency of this stem cell line after subcutaneous injection [47]. The authors suggest that it is crucial, prior to translation into human trials for mandibular reconstruction, to differentiate between the bone formation induced by cells at the site of defect (osteoconduction) and the bone matrix demonstrated by implanted cells (osteogenesis).

## 6. Future Directions in Stem Cell Therapies for Craniomaxillofacial Reconstruction

As we look toward the future of cell-based therapies in regenerative surgery of the craniomaxillofacial skeleton, there is great potential for the expansion of treatments to be translated from preclinical animal models into human clinical trials (see Figure 3). Particularly as targeted stem cell therapies are combined with fundamental knowledge of osteogenic molecular signaling pathways and the environmental factors driving new bone formation, the goal of regenerating missing tissue in a precise, patient-centered manner will be achievable. Some barriers to the ultimate success and broad acceptance of cell-based approaches in reconstructive craniofacial surgery do exist (e.g., cost, accessibility, and implementation). For instance, there is a need to harvest a sufficient number of progenitor cells in order to elicit the targeted effects in vivo through environmental stimuli, through which it can be difficult to achieve clinically meaningful results. Along the spectrum of stem cell populations, embryonic stem cells carry much greater capacity to be omnipotent, being able to differentiate into ectoderm-, mesoderm-, or endoderm-derived tissues when properly directed. Given the current ethical landscape of stem cell therapy using such omnipotent cells, there has been tremendous effort in identifying alternate stem cell niches, such as those we discuss above, from pluripotent lines of differentiation. Particularly exciting as this field continues to innovate and progress is the heightened capacity for inducing pluripotent stem cells into specified lineages of interest by selectively turning on gene expression through genetic engineering approaches. As engineers, scientists, and physicians collaborate in the future progress of stem cell therapies to replace missing bone, there is the potential to alleviate the burden of autologous bone grafts from patients and the incurred costs for the healthcare system as a whole. Although a broad spectrum of alloplastic and allogeneic bone substitute materials has been developed over the past several decades, each of these materials continues to wreak havoc on the host, conferring risks of immunogenicity, infection, and other sequelae of the host endogenous foreign body response. The identification and isolation of postnatal pluripotent stem cell sources has been a major essential step forward toward alleviating this burden. Specific to the field of bone tissue engineering, two stem cell lineages have been particularly successful in preclinical and human clinical trials: BMSCs and ADSCs. As the lauded source of skeletal progenitor cells, BMSCs have shown success when seeded on a host of scaffold biomaterials and applied to the craniofacial skeleton. This work continued through the identification of ADSCs as potent stimulants of bone tissue growth in vivo and is further validated by the accessibility of the stromal adipose tissue, the minimally invasive nature and safety of the extraction and isolation of cells, the great quantity of adipose in multiple regions of the body, etc. ADSCs have thus been identified as attractive candidate cells for the basis of the future development of skeletal tissue engineering constructs in combination with bioactive scaffolds and nanoparticles. This concept has been explored in preclinical models of critical-sized calvarial defects, in which ADSCs were seeded on poly (lactic-co-glycolic acid) (PLGA) scaffolds and implanted in vivo in a mouse model. This resulted in new bone formation at the site of implantation comparable to what previous studies had shown using BMSCs and osteoblasts [48]. With the dawn of a new era of exploration for cell-based regenerative therapies, we look forward to a bright future for craniomaxillofacial surgical applications with the easier, safer identification and isolation of pluripotent cell lines to aid in replacing the foundational bone of the face to restore form and function.

## 7. Conclusions

In summary, we believe that the tools and constructs for reconstructive surgery, especially for the craniofacial region, are diverging away from invasive solutions such as autologous grafts and are focusing on different techniques such as stem cells to regenerate needed tissue. Many methods and techniques have been tested, and some suggest yielding results similar to, if not more successful than, those of these past techniques. Among the innovative and less invasive methods for reconstruction are stem cell regenerative techniques that support natural tissue regeneration with stem cells seeded within biodegradable and/or biocompatible scaffolds. The more commonly used cells are MSCs, both bone marrow- and adipose-derived; however, newer studies have shown the success of iPSCs, hDPSCs, SuSC, and hAFSCs and their potential to be valuable additions to the reconstructive resources. Just as important are the different biocompatible materials and scaffolds used in conjunction with these stem cells, especially in regenerating very difficult and challenging areas such as the mandibular region. Furthermore, targeting transcription factors such as RUNX2 and the BMP-signaling pathway can not only speed up the regenerative process but also optimize the quality and the type of tissue regenerated. The spatial and temporal patterns of BMP, Axin-2, and RUNX2 signaling are crucial to the assembly of appropriately positioned and shaped craniomaxillofacial structures, lending good rationale for the modulation of the differentiation and migration of progenitor cells affected in a morphogenetic gradient to restore structural defects of the craniofacial complex.

In this review, the use of stem cells alongside many different scaffolds and their potential to regenerate tissues such as calvarial bone, palatal bone, craniofacial cartilage, and the mandible were explored. These promising results urge a future change in the “gold standard” of reconstructive surgery (autologous bone grafting) to provide a safer, effective, and less invasive solution for patients.

## Figures and Tables

**Figure 1 jcm-09-03307-f001:**
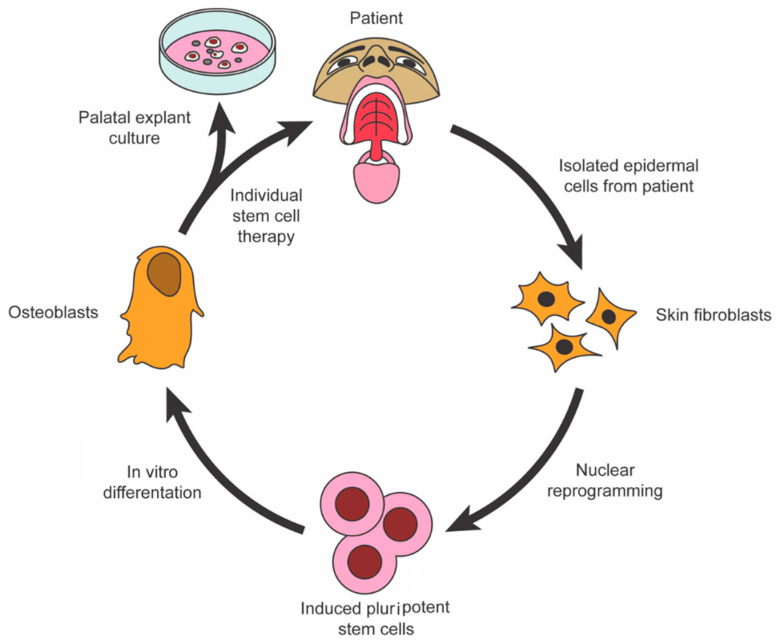
Isolation and programming of induced pluripotent stem cells for use in palatal bone regeneration.

**Figure 2 jcm-09-03307-f002:**
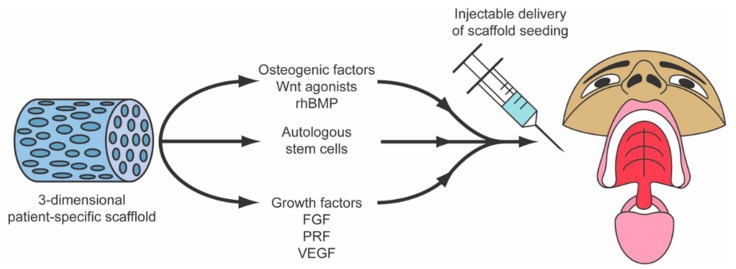
Combined application of osteogenic growth factors and cell-based therapies within biomimetic scaffolds for implantation in the craniofacial region. rhBMP, recombinant human bone morphogenetic protein.

**Figure 3 jcm-09-03307-f003:**
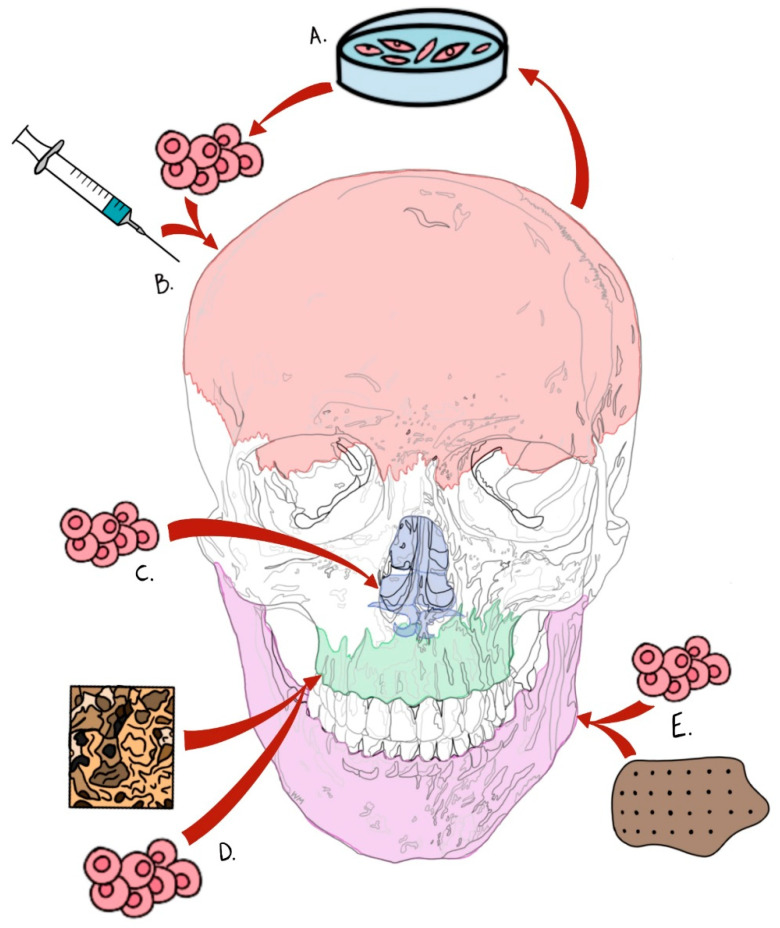
Summary schematic of stem cell-based therapeutics in targeted craniofacial bone tissue engineering. (**A**) Calvarial bone regeneration through engraftment and use of reverted induced pluripotent stem cells (iPSCs), (**B**) delivered via hydrogel injectable system. (**C**) Cartilaginous regeneration within nasal structures via cranial suture-derived or stromal stem cell niches. (**D**) Targeted palatal bone (palatine and maxillary bones) tissue engineering via mesenchymal (adipose and/or bone marrow-derived) stem cell delivery. (**E**) Mandibular defect regeneration via biomaterial polyether ether ketone (PEEK) scaffold delivery of mesenchymal stem cells.

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
