# Peer review of "Stem Cells Regenerating the Craniofacial Skeleton: Current State-Of-The-Art and Future Directions"

_jcm, 2020, doi:10.3390/jcm9103307_

Round 1
Reviewer 1 Report
The review discussed literature on the use of stem cells in craniofacial regenerative therapies. They have focused on various kinds of stem cells in calvarial, palatal bone regeneration, craniofacial cartilage and mandibular regeneration. The stem cells that have been discussed are iPSCs, mesenchymal stem cells, bone marrow, adipose, and tooth-derived stem cells. The use of various scaffolds and factors like Axin-2, Runx2, BMP. 1. Based on the contrast between BMSCs and ADSCs in lines 153 – 155, and the characteristic of ADSCs in line 191, under what circumstances or physiologicial complications will each of them be more favorable for regenerative therapies? 2. In the 3 studies discussed in lines 218-229, undifferentiated ADSCs led to bone regeneration using certain scaffolds (Pourebrahim et 218 al.), while osteogenically-differentiated ADSCs led to bone formation using other scaffolds (Conejero et al.), and cultured ADSCs with palate-derived oral mucosa fibroblasts (Blanco Elices et al.) in conjunction with another scaffold regenerates palatal tissue. Considering the 3 cases, can the authors comment on choosing scaffold materials based on certain characteristics that will be most compatible with the kind/state of cells that are available e.g. undifferentiated, differentiated, cultured, etc. The comments will be helpful because depending on the patient there may be only one kind of cells available at the time of need. 3. The authors mention Runx2, Axin-2, and BMP signaling pathway in facilitating regeneration, which is supported by references. However, these factors orchestrate a controlled spatial and temporal regulation of genes during development, and their own transcription is also temporally and spatially controlled. Based on the applications of these factors as cited in this review, can the authors generally comment on the considerations related to their temporal and spatial regulations that researchers need to be mindful of, when they develop craniofacial regenerative therapies using these 3 factors? 4. The significance of neural crest cells in sections 4 and 5 need to be discussed in brief. 5. In section 3.4. Tooth-Derived Stem Cells, the isolation of multipotent neural crest‐like stem cells from dental pulp needs to be mentioned, with the reference Mehrotra et al. 2019 (PMID: 31738018) 6. Potential issues regarding cost, accessibility, and implementation of stem cell-based craniofacial regenerative therapies need to be briefly discussed. 7. Some sections contain very long paragraphs which need to be split into smaller ones. 8. The following places need references: 8.1 Introduction sentences ending at line 40 and 42 8.2 Section 3.2 sentences contained in lines 157-160, and 161-163 8.3 Section 3.2 line 172 for the current randomized clinical trial, the NCT (National Clinical Trial) number needs to be mentioned 8.4 In section 3.2, during the discussion of biocompatibility of BMSCs, the following reference may be included Biocompatibility of Bone Marrow-Derived Mesenchymal Stem Cells in the Rat Inner Ear following Trans-Tympanic Administration, Journal of Clinical Medcine 2020, 9(6), 1711 8.5 In section 3.2, during the discussion of BMSCs in transplantation, the following reference may be cited. Allogeneic Bone Marrow Mesenchymal Stem Cell Transplantation in Tooth Extractions Sites Ameliorates the Incidence of Osteonecrotic Jaw-Like Lesions in Zoledronic Acid-Treated Rats, Journal of Clinical Medcine 2020, 9(6), 1649. 9. What does the abbreviation SVF mean? 10. Fonts in figures need enlargement.Author Response
- Based on the contrast between BMSCs and ADSCs in lines 153 – 155, and the characteristic of ADSCs in line 191, under what circumstances or physiologicial complications will each of them be more favorable for regenerative therapies? Thank you for your commentary. We discuss specific clinical applications/scenarios in which these respective cell populations will be optimized for regenerative therapies, such as in the following statements found in lines 160-166 of the manuscript text: "BMSCs have been utilized in several alveolar cleft surgeries and show favorable results. In one of these studies, it was concluded that a scaffold-free approach to BMSC bone marrow stem cell-mediated bony reconstruction is safe for alveolar cleft repair, but not indicated for large cleft deficiencies. There have been a number of associations drawn between BMSC populations and their in vivo activity to commercially available demineralized bone matrix (DBM), such as Osteoset DBM, and platelet-derived growth factor when used in tandem with tricalcium phosphate/hydroxyapatite or platelet-rich fibrin composites contributing to bone repair mechanisms." Similarly, we discuss the optimized environments and applications for ADSCs, "Novel culturing methods have also been examined for the packaging and differentiating of human stem cells for palatal bone regeneration, including spheroid delivery systems to optimize cell colonization in scaffold materials in vivo. While still in the initial stages of investigation, these data are promising for the implementation of mesenchymal stem cell adjunctive therapies to induce osteogenesis in cleft defects."
- In the 3 studies discussed in lines 218-229, undifferentiated ADSCs led to bone regeneration using certain scaffolds (Pourebrahim et 218 al.), while osteogenically-differentiated ADSCs led to bone formation using other scaffolds (Conejero et al.), and cultured ADSCs with palate-derived oral mucosa fibroblasts (Blanco Elices et al.) in conjunction with another scaffold regenerates palatal tissue. Considering the 3 cases, can the authors comment on choosing scaffold materials based on certain characteristics that will be most compatible with the kind/state of cells that are available e.g. undifferentiated, differentiated, cultured, etc. The comments will be helpful because depending on the patient there may be only one kind of cells available at the time of need. Thank you for your thoughtful commentary. As these reported outcomes of various scaffold material types in conjunction with cellular therapies are in their infancy of investigation, there is much speculation as to the best scaffold biomaterial/cellular/growth factor therapeutic option in clinical practice. The fundamentals of tissue engineering scaffolds have proven effective in craniofacial tissue engineering (natural biomaterials=more biocompatible, less structural/mechanical stability; synthetic materials=less biocompatible, more structural stability). We have addressed this point in the statement found on lines 232-234 of the manuscript text: "While still in the initial stages of investigation, these data are promising for the implementation of mesenchymal stem cell adjunctive therapies to induce osteogenesis in cleft defects."
- The authors mention Runx2, Axin-2, and BMP signaling pathway in facilitating regeneration, which is supported by references. However, these factors orchestrate a controlled spatial and temporal regulation of genes during development, and their own transcription is also temporally and spatially controlled. Based on the applications of these factors as cited in this review, can the authors generally comment on the considerations related to their temporal and spatial regulations that researchers need to be mindful of, when they develop craniofacial regenerative therapies using these 3 factors? Thank you very much for this thoughtful commentary. We have addressed your comment in full in lines 401-407 of the manuscript text: "...targeting transcription factors such as RUNX2 and the BMP-signaling pathway can not only speed up the regenerative process but also optimize the quality and the type of tissue regenerated. Spatial and temporal patterns of BMP, Axin-2, and RUNX2 signaling are crucial to the assembly of appropriately positioned and shaped craniomaxillofacial structures, lending prudent rationale for the modulation of differentiation and migration of progenitor cells affected in a morphogenetic gradient to restore structural defects of the craniofacial complex."
- The significance of neural crest cells in sections 4 and 5 need to be discussed in brief. Thank you, we have added a brief description of the importance of neural crests cells at lines 278-280.
- In section 3.4. Tooth-Derived Stem Cells, the isolation of multipotent neural crest‐like stem cells from dental pulp needs to be mentioned, with the reference Mehrotra et al. 2019 (PMID: 31738018). Thank you for your suggestion. We have added the reference you provided with a proper description in lines 239-241 in the manuscript text, section 3.4, as prescribed.
- Potential issues regarding cost, accessibility, and implementation of stem cell-based craniofacial regenerative therapies need to be briefly discussed. Thank you for your commentary. We completely agree that these issues must be addressed. We have added a discussion of these important issues in the penultimate section of the manuscript, lines 353-360.
- Some sections contain very long paragraphs which need to be split into smaller ones. Thank you, we have made adjustments to the manuscript during revisions which have mitigated some of these longer paragraphs.
- The following places need references: 8.1 Introduction sentences ending at line 40 and 42 8.2 Section 3.2 sentences contained in lines 157-160, and 161-163 8.3 Section 3.2 line 172 for the current randomized clinical trial, the NCT (National Clinical Trial) number needs to be mentioned 8.4 In section 3.2, during the discussion of biocompatibility of BMSCs, the following reference may be included Biocompatibility of Bone Marrow-Derived Mesenchymal Stem Cells in the Rat Inner Ear following Trans-Tympanic Administration, Journal of Clinical Medcine 2020, 9(6), 1711 8.5 In section 3.2, during the discussion of BMSCs in transplantation, the following reference may be cited. Allogeneic Bone Marrow Mesenchymal Stem Cell Transplantation in Tooth Extractions Sites Ameliorates the Incidence of Osteonecrotic Jaw-Like Lesions in Zoledronic Acid-Treated Rats, Journal of Clinical Medcine 2020, 9(6), 1649. Thank you very much for your suggestions. We have referenced the prescribed JCM articles at the sections you highlighted in the mansucript.
- What does the abbreviation SVF mean? Thank you for pointing this out. We have defined this abbreviation at first mention in the manuscript text (stromal vascular fraction)
Reviewer 2 Report
This review by Jeremie D. Oliver is well organized and accurately describes stem cell regeneration of the craniofacial skeleton.
The following comments should be take into account:
- Please correct the reference style according to the Instructions for Authors.
- Is ASCs in line 180, 190, etc. clearly distinguished from ADSCs?
- It is well explained that various stem cells are used. Please add information about the differences come from stem cells types in craniofacial skeleton regeneration.
- If possible, please mention the modeling technique.
Author Response
1. Please correct the reference style according to the Instructions for Authors.
Thank you for your comment and suggestion. We have updated our in-text references and bibliography at the bottom of the manuscript according to the Instructions for Authors.
2. Is ASCs in line 180, 190, etc. clearly distinguished from ADSCs?
Thank you for this valuable comment. ADSCs is the proper and intended abbreviation for adipose-derived stem cells. We have updated and corrected this throughout the manuscript text.
3. It is well explained that various stem cells are used. Please add information about the differences come from stem cells types in craniofacial skeleton regeneration.
Thank you for your comment and critique. We have addressed your statement through the following statement in lines 141-143 of the manuscript text, referring to the superior osteogenic tissue engineering outcomes through the use of BMSCs, "Compared to other mesenchymal stem cells, stem cells from bone marrow demonstrate increased osteogenic and chondrogenic differentiation potential."
4. If possible, please mention the modeling technique.
While the studies reported in this manuscript are all clinical and translational studies, there were no data discussed on the preclinical modeling of stem cell constructs.
Reviewer 3 Report
Jeremie D. Oliverber et AL., “Stem Cells Regenerating the Craniofacial Skeleton: Current State-of-the-Art and Future Directions” is very interesting and significant for clinicians as well as researchers in the filed of craniofacial regenerative medicine. This review is summarized very well.
Author Response
Thank you very much for your kind and thorough review!